# Effectiveness and Cost-Effectiveness of Non-Pharmacological Interventions among Chinese Adults with Prediabetes: A Protocol for Network Meta-Analysis and CHIME-Modeled Cost-Effectiveness Analysis

**DOI:** 10.3390/ijerph19031622

**Published:** 2022-01-31

**Authors:** Yue Yin, Yusi Tu, Mingye Zhao, Wenxi Tang

**Affiliations:** 1Department of Pharmacoeconomics, School of International Pharmaceutical Business, China Pharmaceutical University, Nanjing 211198, China; yiny9909@163.com (Y.Y.); ttuys0902@163.com (Y.T.); 3220040596@stu.cpu.edu.cn (M.Z.); 2Center for Pharmacoeconomics and Outcomes Research, China Pharmaceutical University, Nanjing 211198, China

**Keywords:** prediabetes, non-pharmacological interventions, Chinese Hong Kong Integrated Modeling and Evaluation, cost-effectiveness, China

## Abstract

Patients with prediabetes who are at a high risk of progressing to diabetes are recommended early-stage intervention, according to guidelines. Non-pharmacological interventions are effective and cost-effective for glycemic control compared with medicines. We aim to explore which non-pharmacological interventions have the greatest potential effectiveness, cost-effectiveness, and feasibility in community-based diabetes management in China. We will perform a systematic review and network meta-analysis to compare the effectiveness of included non-pharmacological interventions, then use Chinese Hong Kong Integrated Modeling and Evaluation (CHIME) to model the yearly incidence of complications, costs, and health utility for the lifetime. Published studies (only randomized controlled trials (RCTs) and cluster RCTs with at least one study arm of any non-pharmacological intervention) will be retrieved and screened using several databases. Primary outcomes included blood glucose, glycated hemoglobin, incidence of type 2 diabetes mellitus, and achievement of normoglycemia. Health utilities and cost parameters are to be calculated using a societal perspective and integrated into the modified CHIME model to achieve quality-adjusted life-year (QALY) estimates and lifetime costs. QALYs and incremental cost-effectiveness ratio will then be used to determine effectiveness and cost-effectiveness, respectively. Our study findings can inform improved diabetes management in countries with no intervention programs for these patients.

## 1. Introduction

Diabetes, one of the top 10 causes of mortality, remains a growing challenge to public health, causing approximately 11.5% of total global health expenditures, with an estimated prevalence of 10.5% (537 million) among adults aged 20–79 years worldwide [1] (p. 2, 57). China has the largest number of adult patients with diabetes (140.9 million as of 2021) [1] (p. 37). Type 2 diabetes mellitus (T2DM) is the most common type [1] and there is no permanent cure for it [2,3]. Treatments can only slow progression of the disease [4] rather than change the decrease in life expectancy; treatment also involves a costly burden of care. However, as recommended by the World Health Organization (WHO) [5], intervention in prediabetes, a period before diagnosed as T2DM, has shown considerable success in preventing the progression of diabetes [6,7,8,9,10,11,12] and even in conversion to normoglycemia [13]. Prediabetes, also called “intermediate hyperglycemia” [14] and “non-diabetic hyperglycemia” [15], includes one or both of two states: impaired fasting glucose (IFG) and impaired glucose tolerance (IGT). IFG is defined as a fasting plasma glucose (FPG) concentration of ≥6.1 and <7.0 mmol/L, and IGT is defined as an FPG concentration of <7.0 mmol/L and a 2-h plasma glucose (2-h PG) concentration of ≥7.8 and <11.1 mmol/L according to the WHO [16]. The American Diabetes Association (ADA) applies the same thresholds for IGT but uses a different diagnostic criterion for IFG (FPG 5.6–6.9 mmol/L), and also defined glycated hemoglobin (HbA1c) of 5.7%–6.4% as prediabetes [17]. According to the International Diabetes Federation, approximately 10.6% of adults worldwide are estimated to have IGT and 6.2% to have IFG as of 2021 [1] (p. 50). Strategies targeting interventions for community-dwelling patients with prediabetes will not only lead to delayed onset of type 2 diabetes (T2DM) but will also offer considerable health benefits.

At present, treatments for prediabetes can be divided into pharmacological and non-pharmacological interventions. Although prophylactic use of metformin [9,18], rosiglitazone, or valsartan has been proven to be effective in delaying the onset of diabetes [8], non-drug interventions including exercise, healthy diet, and alternative medicine, etc., have an important role in the prevention of diabetes [19,20]. The United States Diabetes Prevention Program conducted by Knowler found that, compared with placebo, lifestyle intervention was significantly more effective than metformin, with a reduction of 58% (95% confidence interval (CI): 48% to 66%) in the incidence of diabetes whereas the reduction with metformin was 31% (95% CI: 17% to 43%) at 2.8 years [9]. The Da Qing IGT and Diabetes Study conducted over a period of 6 years showed that 66.7% of people in the control group developed diabetes whereas 46.0% of people in the group with improved diet and exercise developed diabetes [21]. A study conducted by Færch showed that, compared with the control group, the exercise group experienced a reduction in mean amplitude of glycemic excursions at 26 weeks (−21.4, 95% CI: −34.5 to −5.7) [22]. A study conducted by Li found that intensive lifestyle modification in patients with IGT was cost-effective, with a cost-effectiveness ratio of USD 1500/quality-adjusted life-year (QALY), compared with standard lifestyle recommendations or no intervention [23]. However, current systematic reviews or meta-analyses regarding treatments for prediabetes are mainly focused on the following conditions: (i) one kind of non-drug intervention versus control [24,25,26]; and (ii) drug interventions versus one kind of non-drug intervention [27]. Few studies have assessed different prediabetes treatment strategies in terms of effectiveness and cost-effectiveness.

The Chinese government has implemented the China National Plan for Chronic Diseases Prevention and Treatment in 2012–2015 and the China Middle- and Long-Term Plan for Chronic Diseases Prevention and Treatment (2017–2025) [28] to strengthen the management of chronic diseases. Because of these programs, there was a decrease in the prevalence of newly diagnosed diabetes in 2017 [29]. Although the control rate of diabetes was improved compared with that in 2010, it remained the same as that in 2013 [29], which indicates that management has not achieved the expected effects. Considering this, the application prospects of community-based interventions are enormous; however, the health and cost-effectiveness benefits of including patients with prediabetes in management programs remain uncertain. Thus, the need for related evidence is urgent. In the present study, we aim to explore the effectiveness and cost-effectiveness of diabetes prevention strategies and identify a feasible intervention package to provide evidence for the improvement of chronic disease management in China.

## 2. Methods and Analysis

In this study, we will rank the effectiveness of different interventions using network meta-analysis (NMA) and will perform cost-effectiveness analysis (CEA) using the CHIME (Chinese Hong Kong Integrated Modeling and Evaluation) model with a lifetime study horizon; the study flow chart is shown in Figure 1. We will strictly follow the Preferred Reporting Item for Systematic Review and Meta-Analysis for Network Meta-Analyses (PRISMA-NMA) guidance [30] and Consolidated Health Economic Evaluation Reporting Standards (CHEERS) Statement [31] for the study design of the NMA and CEA, respectively.

The protocol has been registered in the International Prospective Register of Systematic Reviews (PROSPERO) with registration number CRD42021291641.

### 2.1. Inclusion and Exclusion Criteria

The research participants included in the NMA must meet the following criteria.

***Included participants.*** (i) Chinese patients, (ii) male or female sex, (iii) over 18 years old, (iv) diagnosed with prediabetes according to the ADA [17] or Chinese Guideline for the Prevention and Treatment of Type 2 Diabetes Mellitus (2020 edition) [32]. The diagnostic criteria are as follows: (i) fasting blood glucose (FBG) 100–125 mg/dL (5.6–6.9 mmol/L) or 2-h PG 140–199 mg/dL (7.8–11.0 mmol/L) or HbA1c 5.75–6.4% (39–47 mmol/mol); or (ii) FBG 6.1–6.9 mmol/L or 2-h PG 7.8–11.0 mmol/L.

***Excluded participants.*** Patients with non-diabetic complications are also included, but those with prediabetes owing to other causes (e.g., endocrine disorders, diseases of the pancreas, pregnancy, medication, viral infection) are excluded.

***Interventions.*** All non-pharmacological interventions related to the treatment of T2DM, whether implemented as stand-alone treatment or adjunct to antidiabetic drugs, will be considered in our study. The included studies must be one of the following types: (i) non-drug interventions vs. control; (ii) non-drug interventions vs. drug interventions; or (iii) non-drug interventions plus drug interventions vs. drug interventions (both groups must have received similar drug treatment).

***Non-pharmacological intervention.*** According to the scoping review (see Search Strategies section), we classified the included non-drug interventions into the following eight categories: (i) nutritional therapy; (ii) physical activity; (iii) psychological interventions; (iv) social network interventions [33]; (v) self-management and education; (vi) media-related interventions; (vii) traditional Chinese medicine; and (viii) multidisciplinary interventions. The contents of each intervention are shown in Table 1. Studies combining different interventions (such as receiving nutritional and psychological treatment at the same time) will be excluded if the effects of the intervention cannot be isolated. Because the costs of intervention must be considered in the economic evaluation, we assumed each intervention is implemented according to eight types of health professional or occupation (Table 1). These practitioners aim to implement relevant interventions and supervise patients follow the glucose-reducing plan strictly. For example, fitness coaches’ work is instructing patients to exercise scientifically and developing a habit; self-management planners aim to supply individualized interventions and promote positive lifestyle behaviors; network nurses receive patient feedback through electronics and adapt the original plan to latest condition. Moreover, the included non-pharmacological treatments need report the frequency of intervention and have been carried out under the guidance of one specific person. Studies that do not meet these two criteria will be excluded. There must also be at least two articles for each of the included interventions.

***Pharmacological intervention.*** The scope of pharmacological intervention was set to include commonly used antidiabetic agents recommended in ADA [17] and Chinese guidelines [32], as shown in Table 1.

***Primary outcomes.*** The primary outcomes include blood glucose (FBG, FPG, or 2-h PG), HbA1c, incidence of T2DM, and achievement of normoglycemia. The definition of T2DM or normoglycemia is based on ADA or Chinese guidelines [17,32].

***Secondary outcomes.*** The secondary outcomes include body weight or body mass index (BMI), blood lipids (total cholesterol, triglycerides, low-density lipoprotein, or high-density lipoprotein), blood pressure (systolic blood pressure or diastolic blood pressure), medication adherence (if patients took antidiabetic drugs at baseline), adverse events related to the intervention, incidence of complications (e.g., hypertension, diabetic foot, nephropathy, retinopathy), and health-related quality of life or health utility.

***Study design.*** Randomized controlled trials (RCTs) or cluster RCTs with a minimum 21-days duration and published in Chinese or English language will be included in this review. For crossover designs, only data from the first period will be extracted owing to concerns about carryover effects [34]. Considering that many RCTs investigating traditional Chinese exercise (Tai Chi, Qi Gong) are conducted in China, we will include Chinese trials if they were approved by a local institution and registered in an international database. There are no restrictions regarding the publication date.

### 2.2. Data Sources and Analysis

***Search strategies***. We will search the following databases or search platforms: PubMed, Embase, Web of Science, the Cochrane Library, Chinese National Knowledge Infrastructure (CNKI), Chinese Science and Technology Periodical Database (VIP), and Wanfang database, from inception to 26 November 2021. We will also check the reference lists of included studies and relevant reviews. Two searches will be conducted, a scoping and systematic search. The scoping search will include observational studies with the aim to identify all non-drug interventions using the search strategy of only including limited participants and study designs (Table 2). Two independent reviewers will select and summarize eligible non-pharmacological interventions by screening the key words and MeSH terms of retrieved papers (Table 1). Combined with the interventions obtained in this step, we will develop a systematic search strategy to identify the final included studies. We will also search for studies involving economic evaluation to provide relevant data for a subsequent economic model.

***Study selection.*** We will import all results of the database searches into NoteExpress (3.5.0.9054, User: China Pharmaceutical University). After removing duplicates, two reviewers will first screen study titles and abstracts in accordance with the inclusion criteria, and then review the full text of potentially eligible papers. Any disagreements will be resolved in discussion or consultation with a third reviewer.

***Data extraction.*** Two reviewers will independently extract the data using Microsoft Excel, and any disagreements will be resolved in discussion or negotiation with a third reviewer. The following information from the included studies will be documented: research characteristics (title, first author, publication year, study design, randomization method, blinding method, allocation concealment), participant characteristics (sample size, age, sex, race, occupation, diagnostic criteria for prediabetes, duration of prediabetes, complications, medication status), intervention characteristics (type, frequency, duration of treatment and follow-up), and outcomes (primary and secondary outcomes, time points reported). Any missing relevant data will not be included in the meta-analysis if still unavailable after contacting the author. We will describe the impact of missing data in the discussion section.

***Risk of bias assessment.*** The risk of bias for the included RCTs will be evaluated by two reviewers using the version 2 of the Cochrane Risk of Bias Tool (RoB 2), including random sequence generation, allocation concealment, blinding, completeness of outcome data, selective outcome reporting, and other bias. Each domain will be judged as low, unclear, or high risk of bias. Considering that it may be difficult to achieve blinding and allocation concealment in most RCTs involving non-drug interventions, we will make note of these and perform sensitivity analysis. Disputes during the evaluation process will be discussed until consensus is reached. If this cannot be achieved, a third reviewer will intervene and resolve the dispute.

### 2.3. Data Synthesis and Statistical Methods

***Network meta-analysis.*** After the systematic review, we will perform a NMA to combine direct and indirect effects, and compare the effectiveness of different interventions using R software (The R Project for Statistical Computing, Vienna, Austria). The rank and probabilities of each intervention will be shown using surface under the cumulative ranking curve (SUCRA) and ranking plots (rank probability − rank curve). SUCRA is a numeric presentation of the overall ranking and presents a single number, ranging from 0 to 1, associated with each treatment [35]. If the SUCRA value of a certain intervention is close to 1, it is always ranked first and if it is close to 0, it is always ranked last [36].

***Effect sizes.*** The risk difference (RD), risk ratio (RR), and mean difference (MD) will be used for the effect sizes. The risk and mean of each group will also be calculated to provide the basis for subsequent economic evaluation. For dichotomous variables (e.g., incidence of diabetes, incidence of complications), the RR and RD will be calculated, as well as their 95% confidence intervals (CIs). For continuous data (FBG, FPG, 2-h PG, HbA1c, BMI, or blood lipids), the MD will be used and standard deviation will also be reported. A forest graph will be used to show the effect size and 95% CI.

***Assessment of heterogeneity.*** In the NMA, we can only include homogeneous studies. Heterogeneity will be calculated with I^2^ quantification. According to the Cochrane Handbook, the results will be combined using a fixed-effects model when I^2^ ≤ 50%. Otherwise, meta-regression analysis will be performed (based on patients’ age, sex, BMI, and sample size of the study) to explore the source of heterogeneity and its influence on the combined effect. After excluding studies with obvious clinical or methodological heterogeneity, a random-effects model will be used in the meta-analysis.

***Subgroup analysis and sensitivity analysis.*** We will perform subgroup analysis based on participants’ characteristics (age, sex, BMI, complications (with or without hypertension), drinking status, and duration of intervention). If possible, we will conduct sensitivity analysis to evaluate the robustness of the results by excluding studies with a high risk of bias or missing data.

***Assessment of inconsistency.*** There may be both direct and indirect comparisons between different interventions, or multiple indirect comparisons (e.g., A vs. B results can be obtained by A vs. C and B vs. C, as well as A vs. D and B vs. D). Combining different studies requires an assessment of inconsistency across direct and indirect evidence or different indirect results. The node split method will be used to evaluate inconsistencies among the included studies. The results will be considered consistent if there is little difference (*p* < 0.5) between direct and indirect comparisons. Otherwise, an inconsistent model must be further considered [37].

***Publication bias.*** A funnel plot will be used to evaluate publication bias when more than 10 studies are included in the NMA [38].

### 2.4. Cost-Effectiveness Analysis

***Study perspective.*** We will apply a perspective of the whole society and consider the direct and indirect costs of patients.

#### 2.4.1. Model Design

The CHIME model was chosen for its strong robustness in predicting the yearly incidence rate over a long period. We also chose the CHIME model because it is used for individual-level discrete-time simulation [39]. The model structure is shown in Figure 2. The cycle length of our model is 1 year, with a lifetime study horizon.

#### 2.4.2. Model Input

***Effectiveness.*** Life-years and QALYs, which can be simulated using the CHIME model, will be used to judge the effectiveness of each intervention.

***Utility values.*** Utility values for each comorbidity and complication are from high-quality studies conducted in China to ensure the reliability and extrapolation of the model.

***Transition probability.*** The yearly transition probability between most states is derived from the CHIME model whereas recurrence of myocardial infarction, recurrence of stroke, and hypoglycemia will be obtained from high-quality literature.

***Natural mortality.*** Natural mortality will be obtained from WHO relevant reports or the Chinese Statistical Yearbook.

***Cost.*** All direct and indirect costs will be calculated, including non-pharmacological intervention costs, drug costs, inspection costs, surgical fees, material costs, treatment fees, diagnosis fees, physician service fees, hospitalization fees, etc. Non-pharmacological intervention costs include the cost of training the professional (Table 1) and of implementing the intervention, which will be obtained from the literature and expert consultation or calculated by weighting the prices of medical services using the Chinese public pricing system. Indirect cost will be calculated based on the human resources approach, that is, the number of days lost × GDP per capita/365. The cost of each intervention will be calculated as the sum of the above costs and will be adjusted to 2022 U.S. dollars.

***Discounting.*** We will adjust the costs and outcomes by discounting the cost and QALYs to 2021 at a 5% (range, 0%–8%) discount rate according to China Pharmacoeconomic Evaluation Guidelines 2019 [40].

#### 2.4.3. Economic Decision

***Decision indicator.*** ICER is calculated using the following formula:$$ICER=\frac{△Cost\;of\;Intervention\;B\;than\;Intervention\;A}{△QALYs\;of\;Intervention\;B\;than\;Intervention\;A}$$

***Threshold.*** We will choose 1–3 times GDP per capita in China in 2022 as the threshold, which is recommended by China Pharmacoeconomic Evaluation Guidelines 2019 [40]. Specifically, ICER lower than 1 times GDP per capita is considered completely cost-effective, ICER between 1–3 times GDP per capita is considered cost-effectiveness with a certain possibility, ICER more than 3 times GDP per capita means with no cost-effectiveness.

#### 2.4.4. Dealing with Uncertainty

***Model assumptions.*** The model structure will follow the CHIME model, an economic model of diabetes, and the model will be validated by Chinese clinical experts.

***Parameter sensitivity analysis.*** Considering the uncertainty of the results when the parameters change into account, we will conduct one-way sensitivity analysis and probability sensitivity analysis. In the one-way sensitivity analysis, we will use the 95% CI of each parameter as the fluctuation interval. For parameters lacking variance information, we assume the relevant parameter fluctuates by 5%–20% (considering the large cost uncertainty), and the discount rate will range from 0% to 8%. The results of one-way sensitivity analysis will be presented in a tornado diagram. For probability sensitivity analysis, prior distribution of the parameters will be applied, such as a beta, normal, and uniform distribution for transfer probability, effect value, mortality, and a gamma distribution for costs. Monte Carlo cohort will be used to simulate 10,000 times, and the cost-effectiveness acceptance curve (CEAC) and incremental cost-effectiveness scatter plot were drawn to display the analysis results.

## 3. Discussion

The proposed research is the first to investigate the effectiveness and efficiency of all non-pharmacological interventions for patients with prediabetes. On the one hand, our study can provide evidence for countries with no intervention programs for patients with prediabetes to support the need for and effectiveness of non-drug interventions. On the other hand, we will compare different non-drug interventions and rank them in an NMA and perform an economic evaluation in this study, which can help local health systems to more intuitively judge the feasibility of a possible intervention package. The method adopted in this study can be referenced by the studies of other chronic disease.

There are some methodological advantages of this study. The search strategy in the study is comprehensive, and we consider the feasibility of interventions in community-based application scenarios. Additionally, our study will focus on the Chinese population; in comparison with the commonly used CORE Model, Cardiff Model and COMT Mode models [41,42,43], it is more accurate to predict the long-term effects of intervention using the CHIME model, which was developed based on a Chinese population. Furthermore, we will extrapolate the results over a long period to comprehensively estimate the cost of non-pharmacological interventions for patients with prediabetes. The results are expected to be more consistent with the reality in China.

However, our study still has some limitations. First, the quality of the literature included in our research will vary. We will ensure the quality of included studies by implementing strict inclusion criteria when screening articles and using sensitivity analysis, meta-regression, subgroup analysis, and other methods to analyze and explain any heterogeneity and inconsistency in the included literature. Second, the classification of different non-pharmacological interventions will be quite difficult in this study. We assume that the management of patients with prediabetes requires a group of specialized professionals, and therefore we will divide non-pharmacological interventions into eight categories according to eight types of health professional or occupation. However, the best way to unify the costs of different non-pharmacological interventions will also be an important challenge in this study.

## 4. Conclusions

In its recent published Standards of Medical Care in Diabetes 2022 [44], the ADA recommends a regular screening for both diabetes and prediabetes in population aged 35 and over. This could bring health benefits and in the same time cause burden for chronic disease management. Our study aims to explore the most cost-effectiveness and feasible intervention package through conducting a systematic review, NMA and economic evaluation on non-pharmacological interventions for prediabetes. The finding of our study will not only help to improve the chronic disease management, but also provide a template for evidence - pooling for decision making process.

## Figures and Tables

**Figure 1 ijerph-19-01622-f001:**
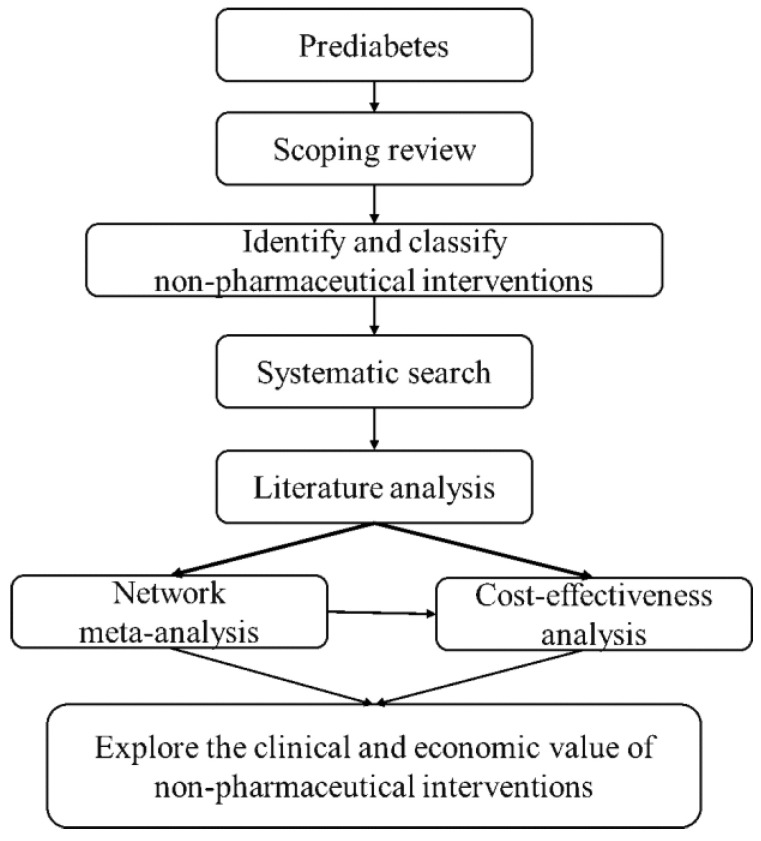
Study flow chart.

**Figure 2 ijerph-19-01622-f002:**
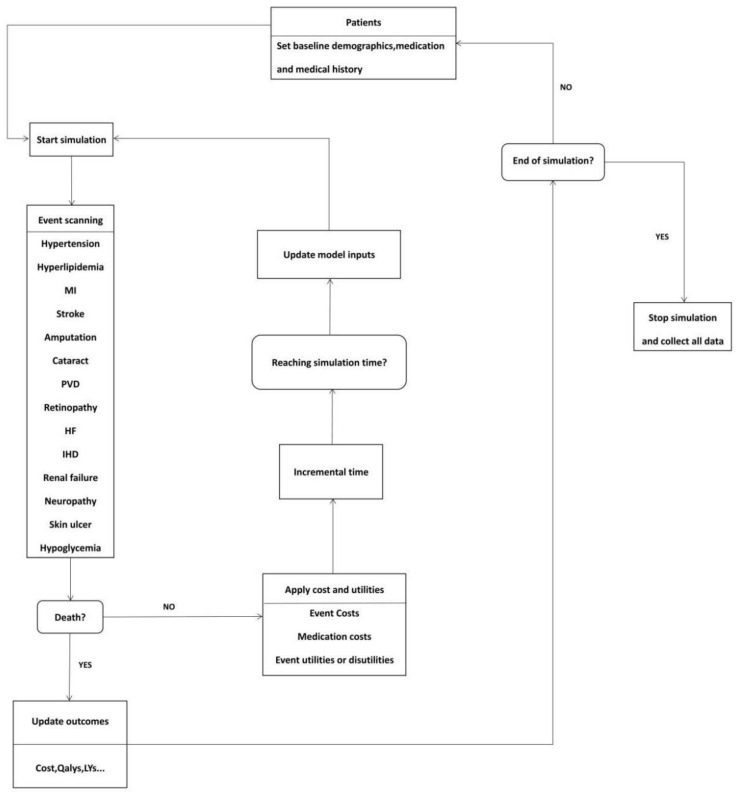
CHIME economic model. (MI, myocardial infarction; PVD, polyvascular disease; HF, heart failure, IHD, ischemic heart disease; QALY, quality-adjusted life-year; LY, life-year.)

**Table 1 ijerph-19-01622-t001:** Classification of interventions.

Type of Intervention		Intervention Measures	Occupation
Non-drug therapies	Nutritional therapy	1. diet OR dietary OR supplementation 2. weight loss OR weight reduction 3. smoking cessation 4. alcohol restriction	Nutritionist
	Physical activity	1. exercise OR training OR sport OR practice OR activity 2. cardio OR anaerobic exercise OR resistance therapy OR physical therapy OR kinesiotherapy 3. bicycle OR walking OR swimming OR yoga OR qigong OR tai chi OR dance	Fitness coach
	Psychological interventions	1. psychological intervention 2. mental health OR emotion OR mood OR neuropsychological 3. meditation OR music OR speech therapy OR interview	Psychologist
	Social network interventions	1. peer support 2. society support OR community support OR family support OR friend support	Peer facilitator
	Self-management and education	1. self-management OR self-management group 2. knowledge, attitude/belief, practice 3. health consultation OR health education	Self-management planner
	Media-related interventions	1. telephone OR mobile OR smartphone 2. application OR software OR internet OR online OR technique OR digital 3. message OR e-mail OR wechat 4. telemedicine OR telehealth OR mhealth OR ehealth OR digital health 5. artificial Intelligence	Network nurse
	Chinese medicine	1. traditional Chinese medicine 2. acupuncture OR acupressure OR massage OR guasha	Physiotherapist
	Multidisciplinary interventions	1. patient care team OR general practitioner	General practitioner team
Drug therapies	Metformin		
	Sulfonylureas		
	Thiazolidinedione		
	Alpha-glucosidase inhibitors		
	Dipeptidyl peptidase IV (DPP-4)		
	Sodium-glucose cotransporter 2 inhibitor (SGLT2i)		
	GLP-1 * receptor agonists		

* GLP-1, glucagon-like peptide 1.

**Table 2 ijerph-19-01622-t002:** Scoping search strategy using Embase database.

No.	Search Items
#1	‘impaired glucose tolerance’/exp
#2	prediabetic: ti, ab, kw OR prediabetes: ti, ab, kw
#3	(progress: ti OR conversion: ti OR develop: ti OR delay: ti OR latent: ti OR potential: ti OR prevent: ti OR prevention: ti) AND (diabetes: ti OR diabetic: ti OR t2dm:ti OR t2d:ti OR niddm: ti)
#4	‘glucose intolerance’/exp
#5	glucose: ti, ab, kw AND (intolerance: ti, ab, kw OR intolerances: ti, ab, kw OR dysregulation: ti, ab, kw)
#6	impaired: ti, ab, kw AND glucose: ti, ab, kw AND (tolerance: ti, ab, kw OR tolerances: ti, ab, kw OR sensitivity: ti, ab, kw OR metabolism: ti, ab, kw OR regulation: ti, ab, kw)
#7	IGT: ti, ab, kw OR IFG: ti, ab, kw
#8	‘impaired fasting’: ti, ab, kw AND (glucose: ti, ab, kw OR glycaemia: ti, ab, kw)
#9	intermediate: ti, ab, kw AND (hyperglycemia: ti, ab, kw OR ‘glycemic control’: ti, ab, kw)
#10	borderline: ti, ab, kw AND (diabetes: ti, ab, kw OR diabetic: ti, ab, kw OR hba1c: ti, ab, kw OR hyperglycemia: ti, ab, kw OR ‘hemoglobin a1c: ti, ab, kw OR a1c: ti, ab, kw)
#11	impaired AND (fpg: ti, ab, kw OR ‘fasting plasma glucose’: ti, ab, kw OR ‘fasting blood glucose’: ti, ab, kw)
#12	#1 OR #2 OR #3 OR #4 OR #5 OR #6 OR #7 OR #8 OR #9 OR #10 OR #11
#13	pregnancy: ti, ab, kw
#14	T1DM: ti, ab, kw OR (‘type 1: ti, ab, kw AND ‘diabetes’: ti, ab, kw) OR T1D: ti, ab, kw
#15	#12 NOT #13 NOT #14
#16	protocol: ti OR guidelines: ti OR consensus: ti OR case: ti
#17	#15 NOT #16
#18	‘animal’/exp NOT ‘human’/exp
#19	#17 NOT #18
#20	‘influencing factors’: ti, kw OR mechanism: ti, kw OR ‘risk factors’: ti, kw
#21	#19 NOT #20
#22	‘crossover procedure’: de OR ‘double-blind procedure’: de OR ‘randomized controlled trial’: de OR ‘single-blind procedure’: de OR (random* OR factorial* OR crossover* OR cross NEXT/1 over* OR placebo* OR doubl* NEAR/1 blind* OR singl* NEAR/1 blind* OR assign* OR allocat* OR volunteer*): de, ab, ti
#23	‘cohort analysis’/exp OR ‘longitudinal study’/exp OR ‘prospective study’/exp OR ‘follow up’/exp OR cohort*: ab, ti
#24	#22 OR #23
#25	‘ecological study’: ti OR ‘case study’: ti OR ‘case report’: ti OR ‘cross section’: ti OR ‘editorial’: ti OR ‘letter’: ti OR news: ti OR ‘newspaper article’: ti
#26	#24 NOT #25
#27	#21 AND #26

## Data Availability

Not applicable.

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
