# Peer review of "Effectiveness and Cost-Effectiveness of Non-Pharmacological Interventions among Chinese Adults with Prediabetes: A Protocol for Network Meta-Analysis and CHIME-Modeled Cost-Effectiveness Analysis"

_ijerph, 2022, doi:10.3390/ijerph19031622_

Round 1

Reviewer 1 Report

The protocol is well structured, I consider it suitable for publication after having resolved some doubts.

In the abstract, however, it is necessary to point out the systematic review which then gives rise to the meta-analysis.

In the methods, I suggest you declare that you will follow the PRISMA-NMA

There is no "Chinese" criterion in the registered protocol. But: “Participants/population: The research population refers to the research subjects of clinical trials, which are patients with prediabetes aged 18 years and above.”

184 RoB2?

I suggest expanding and arguing the network meta-analysis section with references and statements as follows:
"After the systematic review, we will perform a network meta-analysis to combine direct and indirect effects and compare different interventions at the same time. Each intervention will generate curves of efficacy and then a surface under the cumulative ranking curve (SUCRA) analysis to rank the effectiveness of interventions. SUCRA is a numeric presentation of the overall ranking and presents a single number, ranging from 0 to 1, associated with each treatment. Higher odds in SUCRA mean better results. The quality of the processing will be evaluated with the consistency or inconsistency of the analysis.
(References: http://dx.doi.org/10.1016/j.apmr.2020.06.012 , http://dx.doi.org/10.1016/j.ctcp.2020.101260 ). 

Author Response

Thank you for your helpful comments and constructive suggestions, please see the attachment for our responses.

Reviewer 2 Report

Thank you for the opportunity to review your manuscript - Please see attached document for comments and suggestions

Author Response

Thank you for your helpful comments and constructive suggestions, please see the attachment for our responses.

This manuscript is a resubmission of an earlier submission. The following is a list of the peer review reports and author responses from that submission.